# Self-reported negative outcomes of psilocybin users: A quantitative textual analysis

**Bheatrix Bienemann**[1], **Nina Stamato Ruschel**[1], **Maria Luiza Campos**[1], **Marco Aurélio Negreiros**[1], **Daniel C. Mograbi**[1,2] *

**1** Department of Psychology, Pontifícia Universidade Católica, São Paulo, Brazil, **2** Department of Psychology, King's College London, Institute of Psychiatry, Psychology and Neuroscience, London, England, United Kingdom

* daniel.mograbi@kcl.ac.uk

**Data Availability Statement:** The article analyses data already publicly available at www.erowid.org.

**Funding:** D.C.M. acknowledges funding from the National Research Council (CNPq ref 312370) and

## Abstract

Psilocybin, a substance mainly found in mushrooms of the genus psilocybe, has been historically used for ritualistic, recreational and, more recently, medicinal purposes. The scientific literature suggests low toxicity, low risk of addiction, overdose, or other causes of injury commonly caused by substances of abuse, with growing interest in the use of this substance for conditions such as treatment-resistant depression. However, the presence of negative outcomes linked to psilocybin use is not clear yet. The objective of this study is to investigate the negative effects of psilocybin consumption, according to the users' own perception through self-reports extracted from an online platform. 346 reports were analyzed with the assistance of the IRAMUTEQ textual analysis software, adopting the procedures of Descending Hierarchical Classification, Correspondence Factor Analysis and Specificities Analysis. The text segments were grouped in 4 main clusters, describing thinking distortions, emergencies, perceptual alterations and the administration of the substance. Bad trips were more frequent in female users, being associated with thinking distortions. The use of multiple doses of psilocybin in the same session or its combination with other substances was linked to the occurrence of long-term negative outcomes, while the use of mushrooms in single high doses was linked to medical emergencies. These results can be useful for a better understanding of the effects of psilocybin use, guiding harm-reduction initiatives.

## Introduction

The growing use of psychedelic substances has been prominent in epidemiological research. According to the United Nations Office on Drugs and Crimes 2019 World Drug Report, there is an upward trend in recent years on quantities of hallucinogenic substances seized all over the world. This is in agreement with reported qualitative information on increasing use of this class of substances recently [1, but see 2]. Data from the 2019 Global Drug Survey indicates that among the 20 drugs used most prominently over the past year, 6 were psychedelic drugs [3]. From 2017 to 2019, "magic mushrooms" (mushrooms from the genus *psilocybe*) in particular had increases in lifetime use from 24.4% to 34.2% and use in the last 12 months from 10.4% to 14.8% [3, 4]. These increases are mirrored by the growth of the new psychoactive

the Carlos Chagas Filho Research Support Foundation (FAPERJ ref 226501). The funders had no role in study design, data collection and analysis, decision to publish, or preparation of the manuscript.

**Competing interests:** The authors have declared that no competing interests exist.

substances (NPS) market in Europe in the last years [2], with some NPS attempting to mimic the effects of classic psychedelics.

Psilocybin (4-phosphoryloxy-N,N-dimethyltryptamine), the active ingredient in "magic mushrooms", has been investigated in relation to its medicinal properties, in particular for conditions such as treatment-resistant depression (TRD) [5], with suggestions that psychedelic research may lead to a paradigm shift in psychiatry [6, 7]. Psilocybin has also shown potential clinical benefits for depression and anxiety in end-stage cancer [8], possibly with reductions in death anxiety underpinning its therapeutic effects [9]. Although psilocybin is considered a toxicologically safe substance [10–12], there is no scientific consensus on the risks that the use of psilocybin may bring [13].

In a recent study by Carbonaro et al. [14], 10.7% of users reported that, under psilocybin, they placed themselves or others at risk of physical damage; 2.6% reported being violent or physically aggressive with themselves or others, and 2.7% reported having sought help in a hospital or emergency room. Regarding mental health outcomes, significant associations between the consumption of hallucinogens throughout life and mood, anxiety, personality, eating and substance abuse disorders were found in an epidemiological study [15]. This is in agreement with anecdotal evidence indicating persistent anxiety disorder after consumption of mushrooms containing psilocybin [16].

However, there are divergences relative to these findings. In a populational study by Krebs and Johansen [17], no negative mental health outcomes related to the use of classical psychedelics [LSD (lysergic acid diethylamide), psilocybin, mescaline or peyote (*Lophophora williamsi*)] were found. In fact, the authors reported findings indicated that the use of psychedelics was associated with decreased mental health problems. Similarly, another large epidemiological study found no relationship between psychedelic use and incidence of psychosis [18]. In addition, some recent studies have demonstrated the potential for psilocybin to treat or alleviate symptoms present in different clinical conditions [e.g. 6,19–21].

The analysis of self-reported user data is a method often neglected in the scientific literature. There are sites exclusively devoted to the storage and dissemination of information about psychoactive substances, with users visiting these sites to informally publish and share reports of their own experiences with different substances and the outcomes they cause. In addition to serving, potentially, to harm reduction purposes, providing access to information for users, these sites create an opportunity for real-time evaluation of emerging drug trends [e.g. 22].

Psilocybin is capable of promoting intense perceptual changes that include hallucinations, synesthesia, and alterations in temporal perception, as well as changes in emotion and thoughts, which may lead to risk of harmful use [23]. In addition, healthy individuals may experience episodes of *bad trips*–negative experiences, which may involve mental confusion, agitation, extreme anxiety, fear and psychotic episodes–including bizarre and frightening images, severe paranoia, and loss of sense of reality [24]. The relationship between *bad trip* episodes and certain mental states and/or physical *settings* is also relevant to consider subjective aspects as important triggers of anxiogenic outcomes related to the use of psychedelic substances. Understanding the specific circumstances in which psilocybin may lead to negative outcomes may have important implications for the future clinical use of this substance, also providing relevant information for harm reduction initiatives.

Considering this, as well as the scarcity of quantitative analyses of self-reported user data, the aim of this work was to investigate negative effects resulting from the consumption of psilocybin, according to the perception of users themselves. Specifically, we sought to investigate the occurrence of health problems caused by the consumption of the substance, negative acute effects and contextual details of the experiences and possible relationships with the negative outcomes.

## Methods

### Extraction of data and construction of textual corpus

The textual data were obtained from reports manually extracted from the EROWID website (www.erowid.org), a database dedicated to reporting on psychoactive substances and documenting actual reports of users. The reports are reviewed before being published and authors are asked to fulfill certain criteria, such as: description of the context in which the experience was performed and of their previous mental states, details of the preparations made for the use, dosage and time information, observations on possible other medications, herbs or supplements used and a description of the physical and mental effects experienced.

The reports are published anonymously, freely accessible and available on the website in several categories [see 25]. In this research we selected the reports of the subcategories "health problems", "bad trips", "train wreck & trip disasters", present in the category described as Mushrooms (Magic Mushrooms; Psilocybin-containing Fungi). Reports describing the use of mushrooms with substances other than psilocybin as the main active ingredient (e.g. *Amanita muscaria*) were not included. The texts (n = 346) were transcribed manually and any grammar or typing errors were corrected. In addition, some symbols were deleted or replaced (e.g., dashes, quotation marks, indents) to enable analysis by the software. The average length of the reports was 1319.5 words.

In addition, the reports were also categorized according to the following variables: the three subcategories mentioned above ("health problems", "bad trips", "train wreck & trip disasters"), presence of other substances besides mushrooms, dosage, route of administration, form of consumption (dried, tea or fresh and pure) and gender of user (one of the few socio-demographic variables consistently available from the reports). For dosage, a binary variable (doses below and above 5g) was created for the analyses, considering what has been described as a high dose with qualitatively different experiences [26]. Missing values in the variables were classified as *null* and classifications different from the ones mentioned above were classified as *other*. To determine the reliability of the analysis, inter-rater reliability was calculated for all categories that have not been previously provided by the website (i.e., presence of other substances besides mushrooms, dosage, route of administration, form of consumption, and gender of user; see S1 Fig for a model of report). Total agreement between raters was 92.3%, with a kappa of .85 (p < .001).

### Data analysis

The participants' answers were initially analyzed qualitatively and freely, in order to generate familiarity with the content. During this stage, the reports were read in detail, one by one, by two members of the research team. Subsequently, the texts were analyzed quantitatively through IRaMuTeQ 0.7 alpha 2 [27] and R 3.1.2. [28]. The analysis was carried out in the textual corpus constructed from the reports and their categorizations, using text segments (TS). TS are divisions of the text, defining the context in which words appear. TS are automatically sized according to the corpus extension; in this study we used the default division provided by Iramuteq (40 words per text segment; please see S1 Fig). We used the procedures of the Descending Hierarchical Analysis (DHA, Reinert Method); Specificities and Correspondence Factor Analysis (CFA). DHA seeks to obtain textual content clusters with specific meanings, resulting from the similarity, association and frequency of their vocabularies. CFA results in a graphical visualization of the proximities, oppositions and tendencies of the text segments (TS) or corpora classes; locating these elements in a Cartesian graph with factors generated from their classifications and allowing graphical visualization of the co-

occurrence between words and the possible communities in which they coalesce [29]. Specificities analysis indicates the index of co-occurrence between the words, i.e. the relationship of the words between them and the communities formed by groups composed of the words that are most associated.

The criteria for inclusion of both words and categories in their respective classes by DHA are a frequency greater than the mean of occurrences in the corpus and a chi-square value with the cluster greater than 3.84. The words of interest (active forms) selected for analysis were adjectives, nouns, pronouns, verbs, adverbs and forms not recognized by the IRaMuTeQ dictionary. In addition, when words presented with other associated forms (e.g., test, testing, tested), the most frequent form was chosen for graphic representation. The chi-square test values indicate how strongly words and categories are associated with their clusters [29]. We also reported Cramer's V, a measure of effect size for the association [30]. To avoid inflation of type I error, $\alpha$ was set at .01.

## Ethical issues

All materials were anonymized, preventing identification of subjects. Considering that data was public, in agreement with national ethics regulation [31], application for ethics committee approval was dispensed [31; p. 2].

## Results

### Descending hierarchical analysis

The analysis by DHA retained 98.4% of the total corpus, a percentage indicated as acceptable for the corpus to be considered for this type of analysis [29]. The corpus was divided into 12,215 TS, of which 12,414 (98.4%) were retained, relating 15,788 words that occurred 453,711 times (mean of occurrence for TS = 33.55). Of these, the active forms formed 11,239 words, with 2,637 words with frequency greater than six. As can be seen in dendrogram form (Fig 1), DHA resulted in four clusters of words. Initially, the clusters were grouped into two distinct branches, one composed only of cluster 4 (28.1% of total forms classified) and the other composed of another branch with cluster 3 (20.1%) in one of the extremities and a grouping of clusters 1 (30.5%) and 2 (21.4%) in the other. For the association between words and clusters (degrees of freedom = 3), considering that the 25 words with highest association in each cluster are reported, Cramer's V indicated medium and, particularly in Clusters 3 and 4, large effect sizes [30].

### Correspondence factor analysis

The CFA carried out in order to visualize the relation between the clusters indicated that the clusters are divided mainly in three large areas, with cluster 1 and class 2 being strongly related to each other (Fig 2). In relation to the previous categories of the reports and other variables of interest, it is observed that cluster 1 was significantly associated with the subcategory bad trips ($\chi^2$ (2) = 53.81; p < .001, V = .39) and more frequently reported by female users ($\chi^2$ (2) = 13.38; p < .001, V = .20); cluster 2 with train wrecks and trip disasters ($\chi^2$ (2) = 155.92; p < .001, V = .67), use of just mushrooms ($\chi^2$ (1) = 15.56; p < .001, V = .21) and single doses ($\chi^2$ (4) = 11.72; p < .001, V = .18) and cluster 4 with health problems ($\chi^2$ (2) = 67.61; p < .001, V = .46), multiple doses in the same session ($\chi^2$ (3) = 12.39; p < .001, V = .19) and consumption of other substances besides mushrooms ($\chi^2$ (1) = 17.22; p < .001, V = .22). Cluster 3 did not relate to any category and there were no other significant associations (Fig 3).

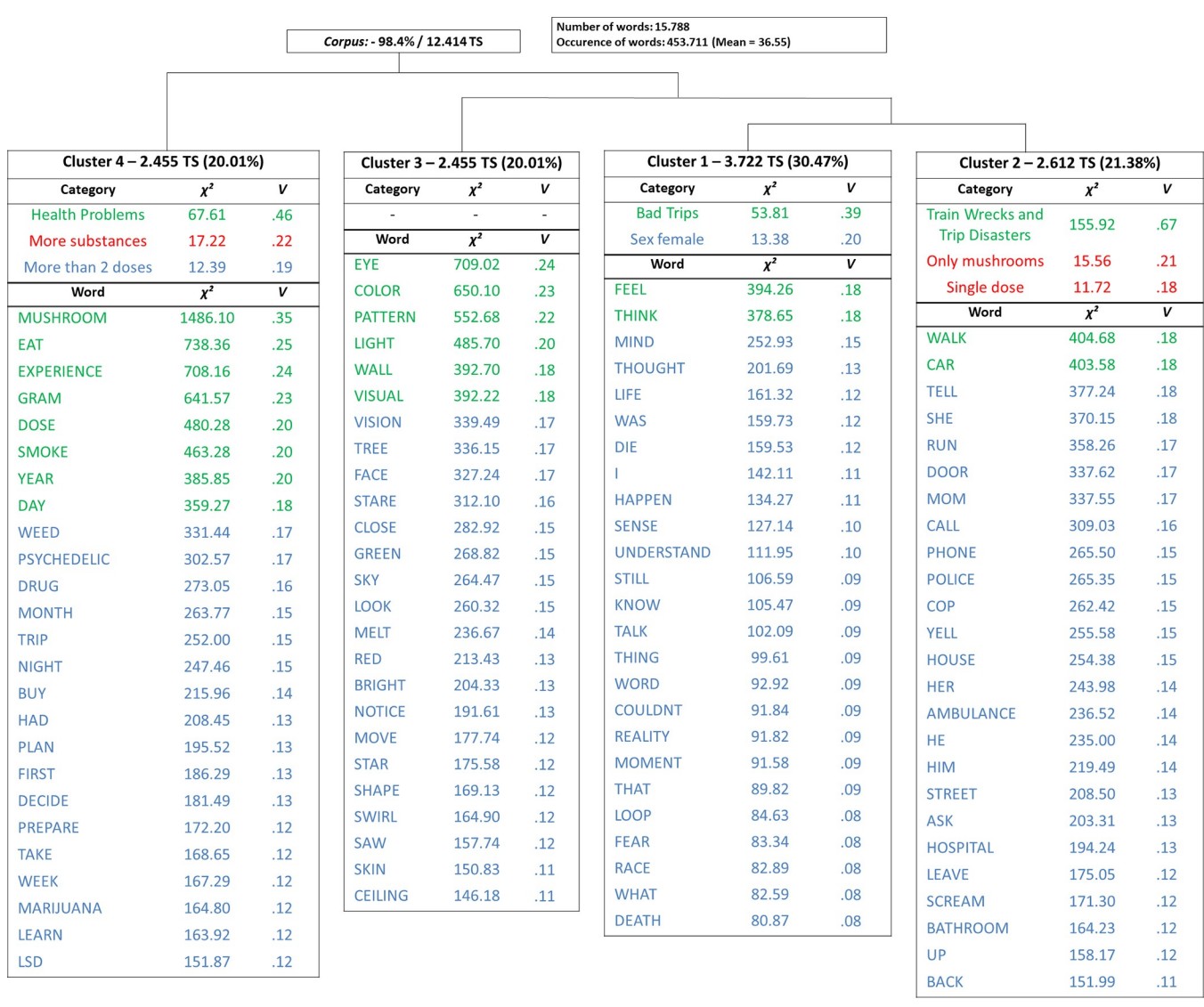

**Fig 1. Dendogram with the 25 words with highest χ² in each cluster.** Small (red), medium (blue) and large (green) effect sizes, according to (30).

## Specificities analysis

The specificities analysis, indicating the index of co-occurrence between the words, can be seen in Fig 4 (cluster1), Fig 5 (cluster2), Fig 6 (cluster 3) and Fig 7 (cluster 4).

## Discussion

The objective of this study was to analyze reports of experiences with psychedelics that led to negative outcomes, according to the perception of the users themselves. The results indicated that the textual corpus was susceptible to this type of analysis. The textual analysis carried out by means of the DHA gave rise to four clusters of words, i.e. four main fields with different meanings in the participants' reports.

Cluster 1, which included 30.5% of TS, has two main axes: "feel" and "think". Although the word "feel" may refer to sensorial experiences, the specificities analysis (Fig 4) indicates that these terms were used in reference to mental elucubrations. This is reinforced by the inclusion

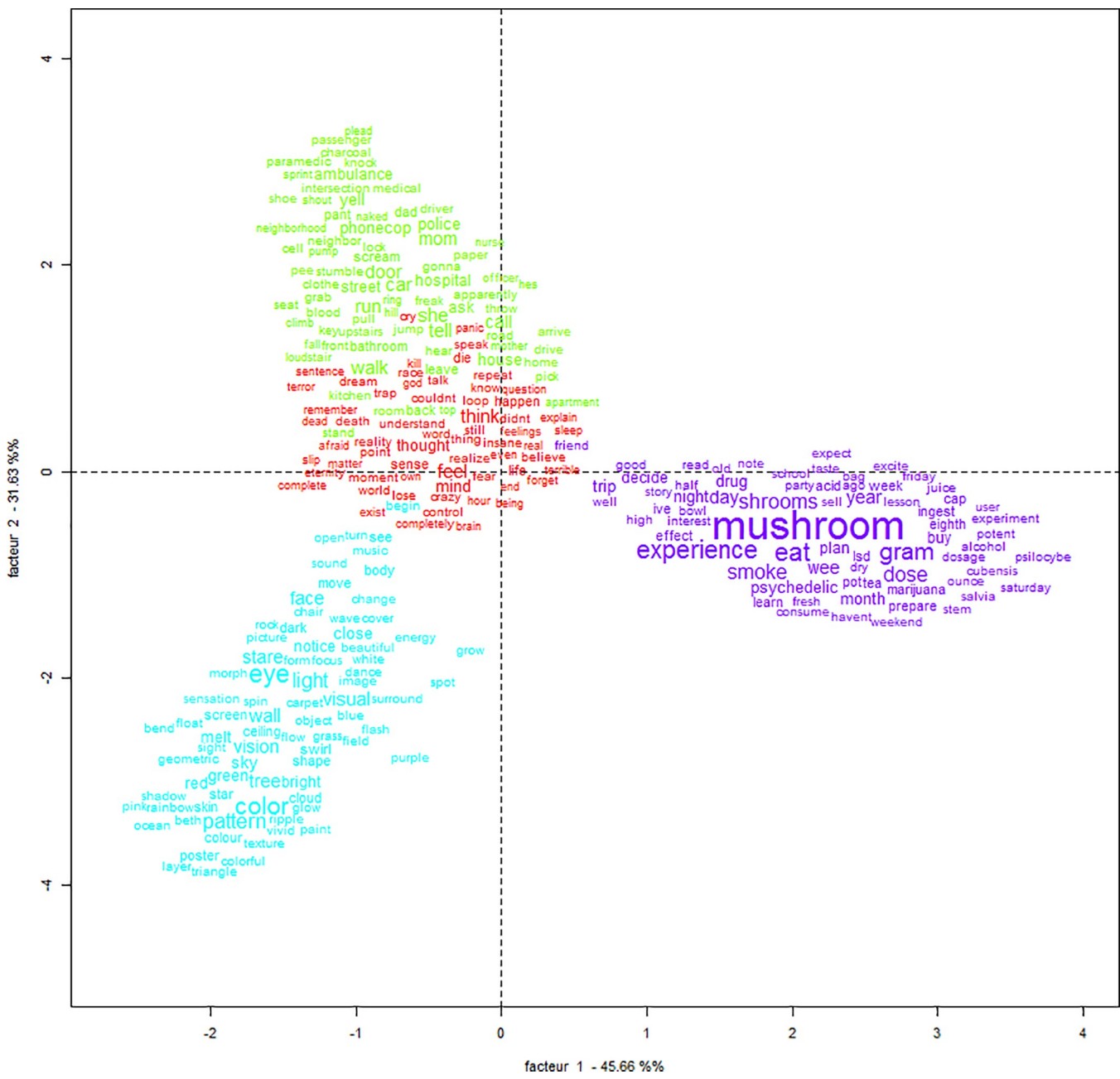

**Fig 2. Relationship between clusters and words in each cluster.** Red–Cluster 1; green–Cluster 2; blue–Cluster 3; purple–Cluster 4.

in the cluster of words and associations such as insane, crazy, mind-race, mind-lose, death, die, fear, among others. The specific contents of this cluster, as well as its association with the category *bad trips*, suggests that short term negative experiences are essentially linked to paranoia and fear/anxiety responses. This is in agreement with previous literature on negative reactions to psychedelics and highlights how these are driven by distortions at the level of thought, co-occurring with anxious states. It suggests that management of anxiety, either by pharmacological or contextual agents (e.g. *setting*) is crucial in the administration of psilocybin. This cluster was also associated with female users. It is possible that this represents stronger effects

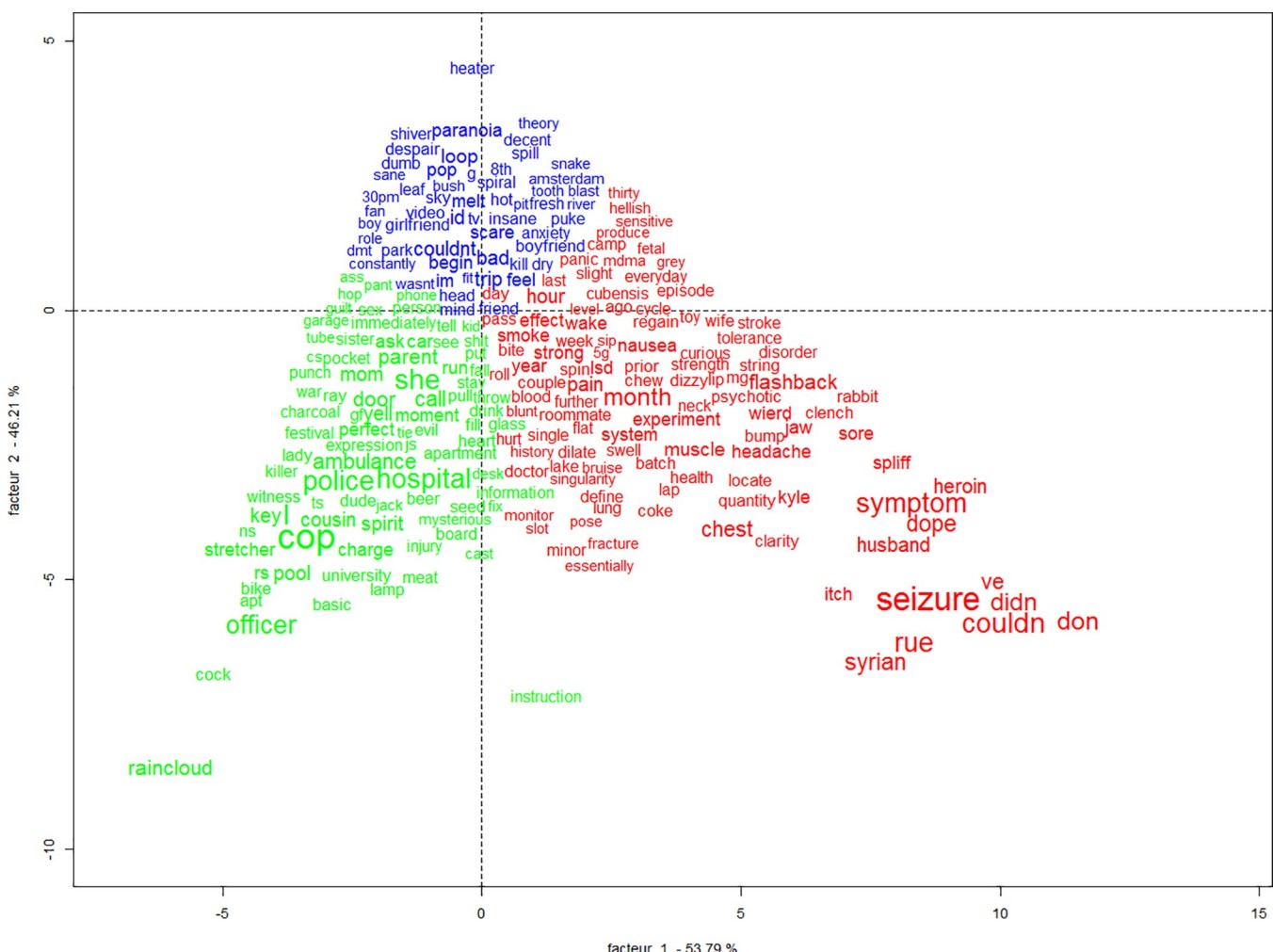

**Fig 3. Words relation by cluster and category associated with each cluster.** Words in blue belong to Cluster 1 that relates to category 3 (Bad Trips), words in green to Cluster 2 that relates to category 2 (Train Wrecks and Trip Disasters), words in red belong to Cluster 4 that relates to category 1 (Health Problems). Cluster 3 did not relate to any category and is not represented in the graph.

in women with similar doses of psilocybin, which could be explained by enzymatic, hormonal or social differences between men and women.

Cluster 2, which accounted for 21.4% of TS, has central words linked to action, e.g. walk, back, tell, call, car. Examining the specificities analysis of this class (Fig 5), indicate that the word associations suggest measures that had to be taken in response to the negative experiences. The presence of words such as ambulance, cop, police, hospital and the significant association of this cluster with the train wrecks and trip disasters subcategory indicate the occurrence of emergencies. Such occurrences probably include the need for medical attention, detention by the police force, need for parental help, etc, also requiring transportation, as indicated by words that refer to the process of getting ready, leaving some place, means of transportation, among others.

This cluster was also associated with single doses of psilocybin only. One way to interpret this result is considering a trend for an association between this cluster and doses above 5g ($\chi^2$ = 4.22; p = .040), which was not significant considering the established $\alpha$ of .01. This suggests that emergencies were linked to single high doses of psilocybin, which is relevant in terms of

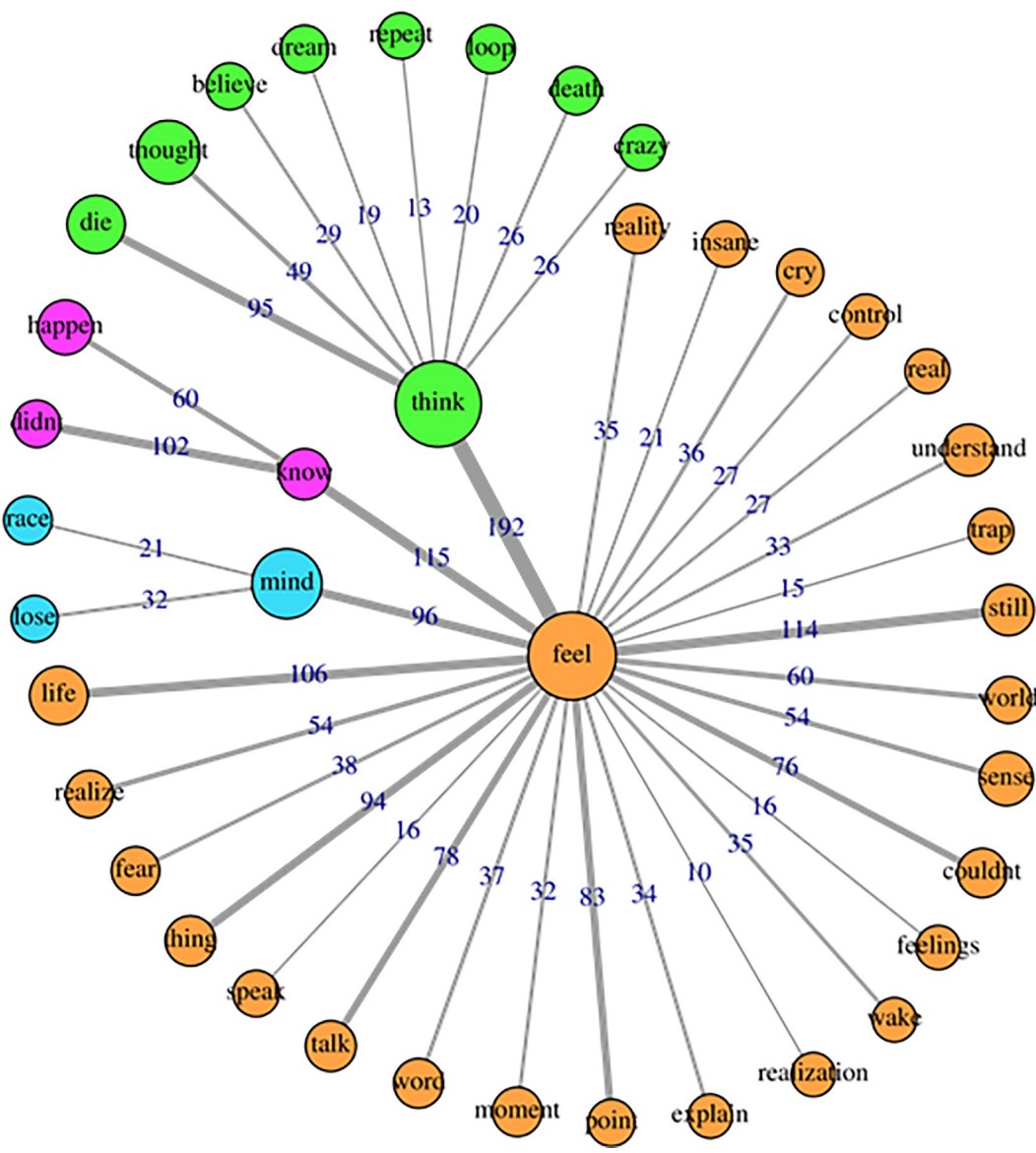

**Fig 4. Co-occurrence and communities for words in cluster 1.**

identifying the safety profile of the substance, also establishing a potential benchmark that may increase unwanted consequences accompanying consumption.

It is important to highlight the strong proximity between clusters 1 and 2, as demonstrated by the CFA (Fig 2). The association between both types of experiences suggests that subjective experiences of bad trips are directly linked to emergencies. The direction of causality, however, is not clear. It is possible that emergency procedures were carried out in response to anxiety (e.g. a request for medical care due to excessive fear of dying or going crazy). Conversely, negative emergence outcomes themselves may have contributed further to the occurrence of bad trips, which are often strongly influenced by the setting in which the experience occurred [13,24,32–35], although this is a less straightforward explanation.

Cluster 3 was made up of 20.1% of TS, with words such as eye, color, pattern, light, visual, vision, stare, referring to visual distortions and sensory-perceptual changes in general, which

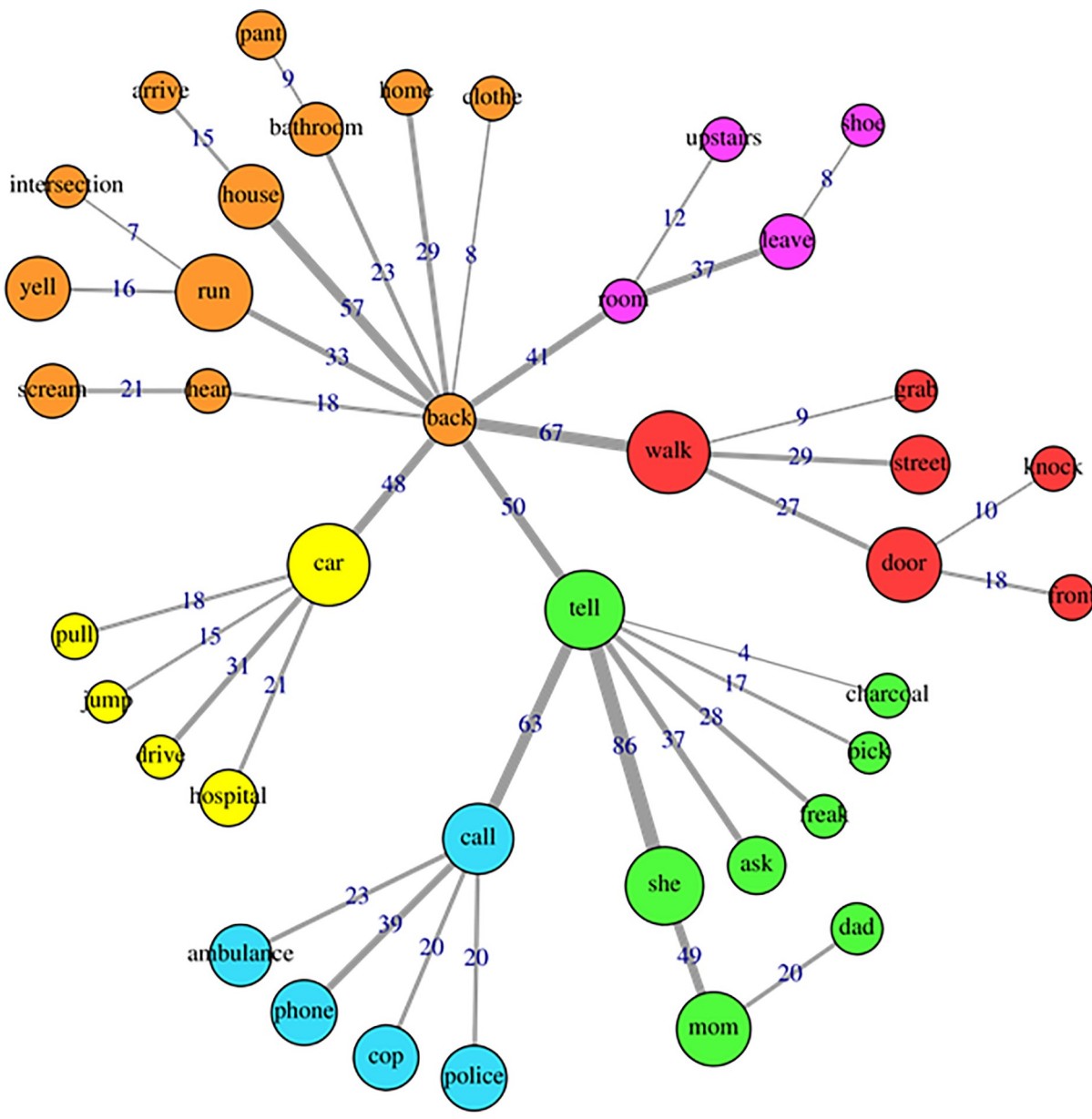

**Fig 5. Co-occurrence and communities for words in cluster 2.**

are well known in psychedelic experiments [14,36–38]. The absence of categories significantly associated with this cluster (Fig 1 and Fig 6) is probably explained by the fact that such described effects are common to psychedelics use as a whole, including benign use.

Finally, cluster 4 collected 28.05% of TS, and agglutinated words that seem to refer to the context of psilocybin use (Fig 7), including preparation of mushrooms (e.g. eat, dry, tea), dosage (e.g. dose, gram, bag), use with other substances (e.g. smoke, weed, LSD) and contextual details such as date (e.g. weekend, month). This cluster was associated with the concurrent use of other substances and use of multiple doses in the same session. In addition, this cluster was also associated with the subcategory "health problems", typically indicative of longer-term complications. It is possible that these complications are consequences of use associated with

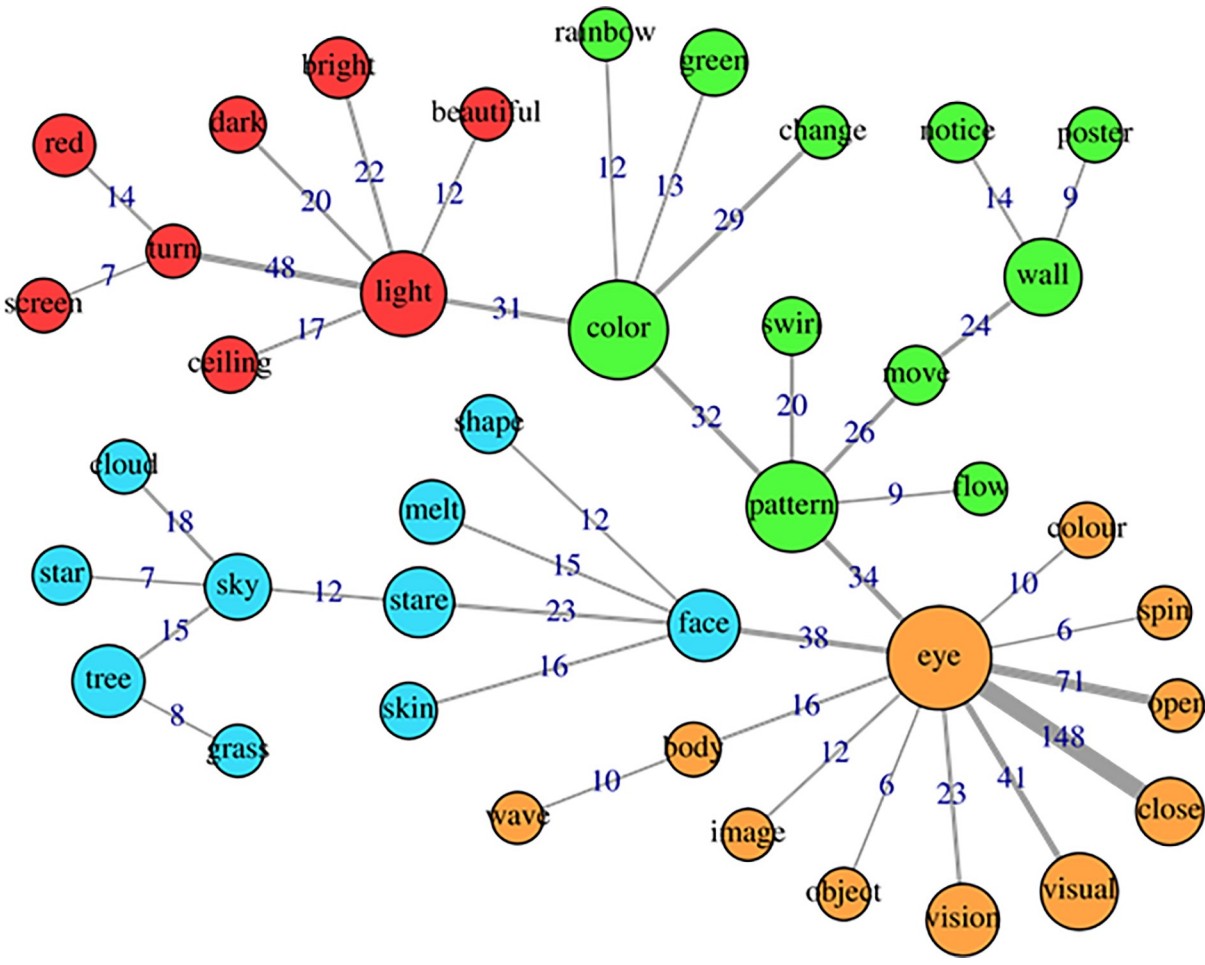

**Fig 6. Co-occurrence and communities for words in cluster 3.**

other substances, given the presence of words like weed, pot, marijuana, LSD, acid. This fact may reinforce the findings about negative outcomes that occur more frequently due to the use of psilocybin associated with other substances, especially alcohol [13,39], and may explain discrepancies in the literature in relation to the association between psychedelic use and negative mental health outcomes.

## Conclusion

This study aimed to analyze self-reports of negative experiences with psilocybin according to the perception of the users themselves. Psilocybin has been used for centuries, with increased medical interest in recent decades, but the wealth of experience of users has rarely been investigated with sound methodology in the scientific literature. To the best of our knowledge, this is the first study to analyze, using appropriate software, the structure and associations of user self-reported experiences. Findings reinforce the need to manage anxiety during psilocybin administration [24], indicating that distortions at the level of thought were the main cause for bad trips. Additionally, these bad trips were also associated with high doses of psilocybin as well as with emergencies. Longer-term health problems were associated with multiple doses and concurrent use with other substances, in agreement with existing literature [13, 39]. These

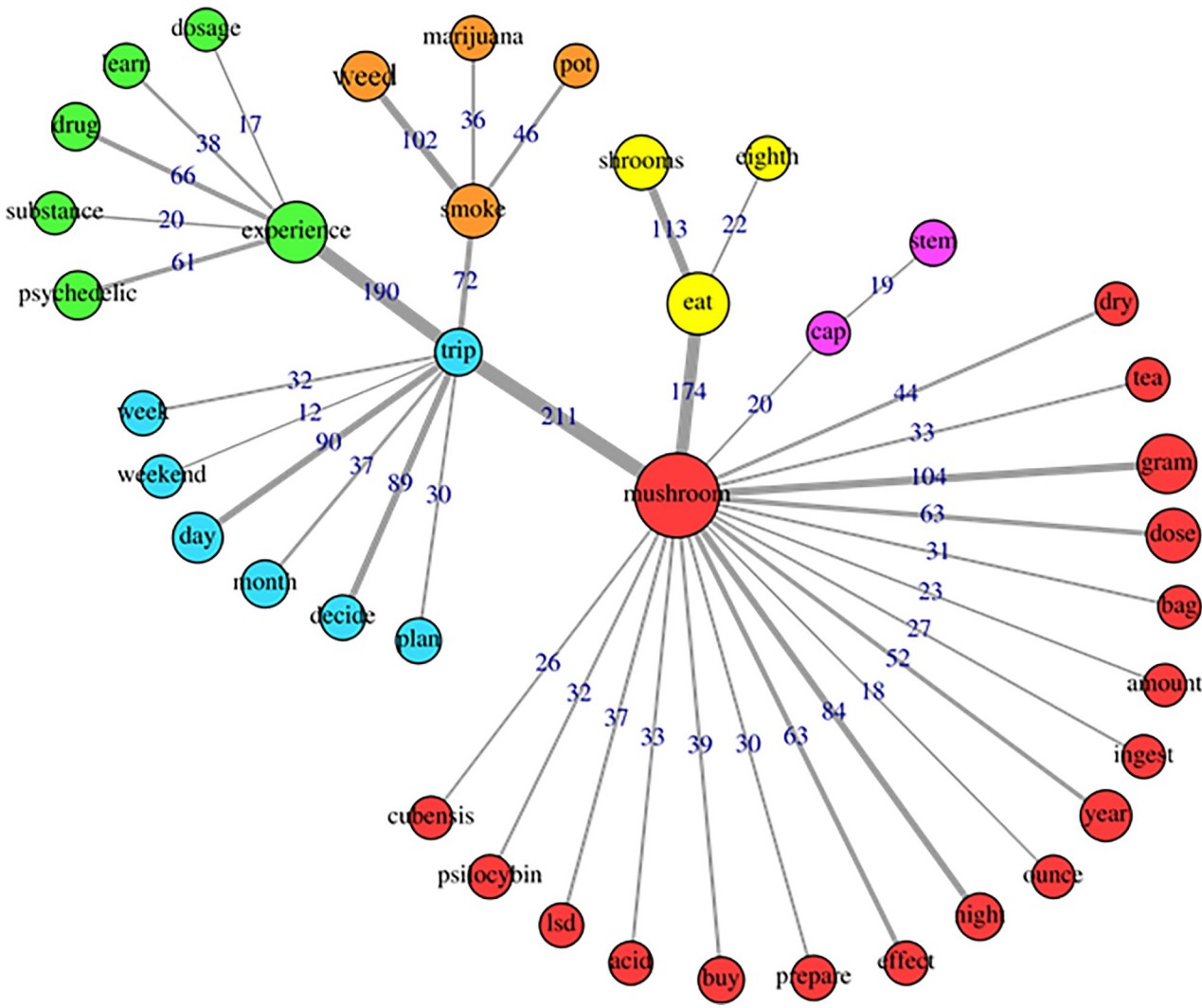

**Fig 7. Co-occurrence and communities for words in cluster 4.**

findings clarify individual and contextual elements that may precipitate negative outcomes linked to psilocybin use, assisting in the elaboration of safety guidelines for users and/or researchers.

The study has a number of important limitations, including a large number of missing values, which prevented the analysis of contextual variables, including setting-specific information. Another limitation refers to sampling, with these reports potentially not representing fully psilocybin users and even with the effective use of substances not being secured in fact, as they only come from reports shared online. This consideration should be done together with the issue of self-selection, that can promote a biased sample with non-probability sampling, considering only experiences that are reported by the users at the website. Additionally, the illegal status of psilocybin is also a potential confounder for results, as the negative outcomes may be connected to black market influences (e.g. different substance being consumed, lack of information about freshness of mushrooms) and not to the substance itself. Nevertheless, given that psilocybin remains being consumed illegally, the current findings provide information valuable to understand use under current circumstances. Finally, the study is exploratory

in nature. In this sense, the current study may be used to generate hypotheses by other researchers in the field conducting experimental work, helping to clarify the relationship between contextual variables and subjective effects of psychedelic experience, including the content of the "trips" reported by the users. Further studies are needed to establish more consistently the long and short term consequences of psilocybin use.

## Supporting information

**S1 Fig.**
(TIF)

## Acknowledgments

The authors acknowledge the work done by Erowid.org in providing information about the use of psychoactive substances and promoting increased awareness on this topic.

## Author Contributions

**Conceptualization:** Bheatrix Bienemann, Daniel C. Mograbi.

**Data curation:** Bheatrix Bienemann, Nina Stamato Ruschel.

**Formal analysis:** Bheatrix Bienemann.

**Investigation:** Bheatrix Bienemann, Nina Stamato Ruschel.

**Methodology:** Bheatrix Bienemann, Daniel C. Mograbi.

**Resources:** Daniel C. Mograbi.

**Supervision:** Marco Aurélio Negreiros, Daniel C. Mograbi.

**Writing – original draft:** Bheatrix Bienemann, Maria Luiza Campos, Daniel C. Mograbi.

**Writing – review & editing:** Nina Stamato Ruschel, Marco Aurélio Negreiros.

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
