## [Decision Letter · Decision Letter 0]

21 Nov 2019

PONE-D-19-25745

Self-reported negative outcomes of psilocybin users: A quantitative textual analysis

PLOS ONE

Dear Daniel,

Thank you for submitting your manuscript to PLOS ONE. After careful consideration, we feel that it has merit but does not fully meet PLOS ONE’s publication criteria as it currently stands. Therefore, we invite you to submit a revised version of the manuscript that addresses the points raised during the review process.

We would appreciate receiving your revised manuscript by 20 February 2020. To enhance the reproducibility of your results, we recommend that if applicable you deposit your laboratory protocols in protocols.io, where a protocol can be assigned its own identifier (DOI) such that it can be cited independently in the future. For instructions see: http://journals.plos.org/plosone/s/submission-guidelines#loc-laboratory-protocols

We look forward to receiving your revised manuscript.

Kind regards,

Giuseppe

Giuseppe Carrà, MD, PhD

Academic Editor

PLOS ONE

Journal Requirements:

D.M. acknowledges funding from FAPERJ and CNPq.

D.C.M. acknowledges funding from the National Research Council (CNPq ref 312370) and the Carlos Chagas Filho Research Support Foundation (FAPERJ ref 226501). The funders had no role in study design, data collection and analysis, decision to publish, or preparation of the manuscript.

Additional Editor Comments:

Dear Daniel,

reviewers have now completed their task. As you'll see the manuscript has merits, though several revisions are needed before it can be considered suitable for publication.

I'd be glad to consider a revised version of your interesting work.

WBR

Giuseppe Carrà

Academic Editor

Reviewers' comments:

Reviewer's Responses to Questions

**Comments to the Author**

1. Is the manuscript technically sound, and do the data support the conclusions?

Reviewer #1: Yes

Reviewer #2: Yes

2. Has the statistical analysis been performed appropriately and rigorously? 

Reviewer #1: Yes

Reviewer #2: Yes

3. Have the authors made all data underlying the findings in their manuscript fully available?

Reviewer #1: Yes

Reviewer #2: No

4. Is the manuscript presented in an intelligible fashion and written in standard English?

Reviewer #1: Yes

Reviewer #2: Yes

5. Review Comments to the Author

Reviewer #1: Comments for Manuscript Number PONE-D-19-25745

Self-reported negative outcomes of psilocybin users: A quantitative textual analysis

This is an interesting study based in an unusual approach which accessed publically available self-reported negative outcomes of psilocybin use.

Research on illicit drugs is largely focused on epidemiological data and on etic more than emic perspectives. To understand people`s experiences with psychoactive substances by their own words is absolutely crucial to produce useful knowledge when it comes to design harm reduction strategies or to address possible clinical potential. This is, therefore, a relevant approach. Besides that, the manuscript is clear for both expert and non-expert publics, well organised and well written.

Authors explain the need for that kind of research approaches and sustain what is said in scientific literature. Nevertheless, I think it is possible and desirable to go much further on references used, since this is a hot topic with increasing stiduies available. For example, to address the growing use of psychedelic substances, authors make reference to the Global Drug Survey. This is valid source but there are others (like UNODC or EMCDDA reports) that can be combined with this one. This is even more important if we take in consideration the levels of uncertainty when it comes to prevalence of illicit drug use and to the low control of biases involved in the global drug report collection of data. Besides that, many studies on the effects of psilocybin are now being published, both in controlled and uncontrolled settings.

The introduction section includes complementary and contradictory data regarding psilocybin potential benefits and harms, which is positive. It also explains how the study will try to contribute to reduce lacunar knowledge in that scenario. A minor comment to that section: at a certain point, authors state that “these sites [devoted to the storage and dissemination of information about psychoactive substances] (…) acting as a valuable resource for prospective or current users”. I tend to agree with this, but we truly don`t know if it is true, unless studies document those positive outcomes…

Methods used and correspondent analysis are adequate to an exploratory study (and the fact that there were two researchers performing the analysis is important) and well described. I personally think that a classic content analysis would bring much more insight over what is the aim of the study but, given the exploratory nature of this approach, this research is already a step forward in producing necessary knowledge.

Minor comments to the method section: authors simply mention that gender of the user will be a variable to analyse, but don´t explain why this was the only socio-demographic element taken in consideration.

However, I see two problems with this methodological strategy:

- one is referred by the authors at the end of the article and refers relevant biases related to the representation of the sample;

- the other one is not mentioned and I think it would be fundamental to take in consideration. Since psilocybin use is classified as illicit, we cannot know, for sure, to what level are those negative outcomes really connected to psilocybin itself or to problems related to the black market vicissitudes (for example not being psilocybin the substance in question). It would be interesting, also, to combine in the conclusion section the results of the study with more scientific literature regarding the same topic, since there are already many studies on the effects of the substance.

Reviewer #2: After reviewing this study, I think it is a solid piece of research. There are, however, a few methodological considerations to be made with the suggestion of providing further information in a revised version of the manuscript.

Before that, I would like to suggest the author that a word of caution should be expressed in the discussion section about the data on which the study is based. They are self-reported reports of the use of psilocybin, and therefore no 'ground truth' about their effective use of substances can be secured. This consideration should be done together with the issue of self-selection

- Having said that, there a few problems with the methodology description that needs clarification:

Reports have been manually categorized using several variables. Some are reporting plain information, while some are more interpretative. Therefore a form of inter-coder reliability should be added since this coding is crucial for the following analysis using Iramuteq. Examples of such coding the appendix would be welcome.

- The automatic text analysis deployed is sound, and the use of DHA is appropriate giving the context, but some additional information should be provided about what was the unit of context (UC) on which the analysis is performed. Later in the text, the authors mention 'text segments', but it is not clear how these were formally defined. This is an important point as the results of the DHA can vary substantially.

Information about the average length of the reports should be included.

-The outcome of the DHA is expressed in chi-square statistics that reveal the presence of association, in this case between a given set of 'structural variables and linguistic patterns. The relative strength of this association is not part of the output of the software itself, but the authors could test it using phi and creamer's v.

If the above points are addressed in a revised version of the manuscript, I believe the paper to be fit for publication.

6. PLOS authors have the option to publish the peer review history of their article (what does this mean?). If published, this will include your full peer review and any attached files.

Reviewer #1: No

Reviewer #2: No

---

## [Author Response · Author response to Decision Letter 0]

26 Jan 2020

Dear Giuseppe,

Thanks for the careful editing of the manuscript. In addition to point-by-point replies to the reviewers (please see the response to reviewers file, at the end of the submission), we updated the files names according to the journal guidelines and removed funding information from the acknowledgements. The current information on the funding statement section is correct.

If you need any additional information, please do not hesitate to contact us.

All the best,

Daniel

---

## [Editor Report · Decision Letter 1]

30 Jan 2020

Self-reported negative outcomes of psilocybin users: A quantitative textual analysis

PONE-D-19-25745R1

Dear Daniel,

We are pleased to inform you that your manuscript has been judged scientifically suitable for publication and will be formally accepted for publication once it complies with all outstanding technical requirements.

With kind regards,

Giuseppe

Giuseppe Carrà, MD, PhD

Academic Editor

PLOS ONE

---

## [Editor Report · Acceptance letter]

5 Feb 2020

PONE-D-19-25745R1 

Self-reported negative outcomes of psilocybin users: A quantitative textual analysis 

Dear Dr. Mograbi:

I am pleased to inform you that your manuscript has been deemed suitable for publication in PLOS ONE. Congratulations! Your manuscript is now with our production department. 

With kind regards,

on behalf of

Dr. Giuseppe Carrà 

Academic Editor

PLOS ONE